# Root Response of Moso Bamboo (*Phyllostachys edulis* (Carrière) J. Houz.) Seedlings to Drought with Different Intensities and Durations

Zhenya Yang [1,2,3], Yonghui Cao [1,3], Jiancheng Zhao [2], Benzhi Zhou [1,3,*], Xiaogai Ge [1,3], Qin Li [2] and Maihe Li [4,5]

1   Research Institute of Subtropical Forestry, Chinese Academy of Forestry, Hangzhou 311400, China; yangzhenya1234@163.com (Z.Y.); fjcyh77@sina.com (Y.C.); gexiaogai2006@163.com (X.G.)
2   Zhejiang Provincial Key Laboratory of Bamboo Research, Zhejiang Academy of Forestry, Hangzhou 310023, China; jiancheng68@163.com (J.Z.); liqin3860@163.com (Q.L.)
3   Qianjiangyuan Forest Ecosystem Research Station, National Forestry and Grassland Administration, Hangzhou 311400, China
4   Forest Dynamics, Swiss Federal Research Institute WSL, CH-8903 Birmensdorf, Switzerland; maihe.li@wsl.ch
5   Key Laboratory of Geographical Processes and Ecological Security in Changbai Mountains, Ministry of Education, School of Geographical Sciences, Northeast Normal University, Changchun 130024, China
*   Correspondence: bzzhou@caf.ac.cn

**Abstract:** The root of Moso bamboo (*Phyllostachys edulis* (Carrière) J. Houz.) develops extremely rapidly at seedling phase and is highly sensitive to water content in soil, but its response patterns and adaptation strategies of its root to drought are little known. The aim of this study was to investigate the response of root morphology and architecture of Moso bamboo to drought at seedling phase and then to explore the drought adaptation strategies of its root. One-year-old potted seedlings of Moso bamboo were planted under three drought treatments (control, moderate drought and severe drought) for three months. Seedling growth, specific root length (SRL), root architecture (fractal dimension (FD), root branching angle (RBA) and root topological index (TI)) and non-structural carbohydrate (NSC) concentrations in roots were measured every month. The results are as follows: (i) The dry weight of root and shoot decreased significantly under drought stress. (ii) The SRL decreased under drought stress in the early duration (the first month), and then increased in the late duration (the third month). Both FD and RBA decreased, while TI and the concentrations of NSCs increased under drought stress. (iii) The NSC concentrations were positively correlated with SRL and TI, but exhibited an inverse relationship to FD and RBA. Our results indicated that Moso bamboo seedlings formed a "steeper, simpler, expensive (low SRL and high TI)" root architecture to adapt to a short-term drought (one month), and formed a "cheaper (high SRL)" root to adapt to a long-term drought (three months). Increase of NSC concentrations supported the root architecture plasticity to some extent.

**Keywords:** drought; adaptation strategy; root architecture; non-structural carbohydrates

## 1. Introduction

Drought is one of the most important agroforestry disasters in China and is a global issue constraining the development of agriculture and forestry [1,2]. Current projections indicate that future droughts are bound to be even more frequent and destructive due to rising temperatures associated with global warming [3–5]. A better understanding of plant adaptation strategy and physiological mechanism under drought is vital for improving management practices in agriculture and forestry for predicting the fate of natural vegetation under climate change.

Roots are capable of responding to drought through a series of adaptation strategies, including root biomass adjustments [6], rooting depth [7], and physiological plasticity [8],

to make plants avoid and tolerate drought stress. It is difficult to make general statements about root growth in response to drought so far. Root length, root surface area, root volume, and root tips have decreased under intense and durative drought stress or increased under short-term moderate drought stress [9,10]. Compared with the changes in the quantity of root, changes in specific root length (SRL) under drought stress are more likely to reflect root formation strategies [11–13]. The carbon utilization efficiency and formation cost of the root system can also be judged by analyzing the SRL. Some woody plants improve water absorption of roots under drought by changing the diameter of the xylem conduits, resulting in changes in root diameter and SRL [7]. Olmo et al. [14] found that SRL increased under drought conditions after studying the drought resistance response of seedlings of 10 woody tree species. This seems to be an advantage in that plants build longer roots with less carbon when water is limited. On the other hand, some studies found that SRL seemed to be especially resistant to drought [7,15]. The effect of drought on SRL has been a subject of controversy so far. Observing the response of roots with different diameters under drought conditions seems to clarify the adaptation strategy of trees, because the SRL and tissue density of roots with different functions show different responses under stress. Thick roots with transport and storage functions tend to be conserved under drought conditions [16,17], while fine roots with absorption function are affected more severely by drought [14]. We speculate that the response of roots to drought may depend on root function, tree species and drought intensity.

The root advantages of adaptable tree species in a long-term arid environment are often reflected in root architecture. Roots with special topology and branching patterns have stronger drought adaptability and higher water absorption capability [18–20]. Plants with "steeper, deeper, cheaper (finer root or root with high SRL)" root architecture [7,21] or a well-developed taproot [15,22] have more advantages in coping with drought. Compared with the fibrous root tree species with developed lateral roots, the taproot tree species with developed taproots tend to be more stable in arid soils [23,24]. It is still unclear how the root architecture changes as an adaptation strategy under drought conditions, although the ideal root architecture for growth in dry soil has been reported. Two extreme root architectures of herringbone branching and dichotomous branching (Figure 1) have been reported based on topology [25]. Dichotomous branching roots often show strong competitiveness in resource-scarce soil because they can quickly occupy soil space in resource-scarce soil through branching rapidly, with lower carbon cost. In contrast, herringbone branching roots absorb water more effectively in deep soil layers than shallow root systems because their taproots are developed and require more carbon [11,15,24–27]. With the introduction of fractal theory, the complexity of plant roots can be highly quantified. The idea of fractal has been discussed since the 19th century and then it was not gradually considered to be able to estimate and quantify the complexity of form, shape or texture of objects until 1977 [28]. The fractal dimension (FD) of the plant root system can reflect the complexity of root branching accurately and quantitatively under different environmental conditions [28,29]. It is generally recognized that the higher the FD is, the more complex the root system will be [29–31]. Variations in the FD of roots in response to genotype and nutrient supply, except drought, have been reported in a number of species [28,32]. Despite the increasing body of literature on the impacts of water deficit on seedlings and forest ecosystems, few experimental studies have evaluated the change in root architecture quantitatively under drought condition by topology and fractal.

Exploring the distribution of photosynthetic products under drought conditions seems to be a more direct way of studying root responses to drought from the perspective of carbon investment. Non-structural carbohydrates (NSCs, i.e., mobile sugar and starch) are important photosynthesis products supporting plant growth, metabolism and a series of physiological activities [33,34]. The concentration of NSCs strongly affects plant growth and is sensitive to fluctuant environmental factors, such as nutrients, water and atmospheric carbon dioxide [35–38]. Change in the NSC concentration at growth parts can improve the flexibility of the plant growth in response to fluctuating environments [39–41]. Active

parts of root proliferation tend to accumulate more NSCs [42]. The concentrations of NSCs in roots are closely related to root proliferation [39]. However, the change in NSC concentration has not been considered as an adaptation strategy related to root plasticity. In general, the accumulation of NSCs, SRL and root topological index (TI) can reflect the root formation strategy and the carbon input cost. Studies have reported a balance between NSC concentrations and root radial growth as well as the formation of lateral roots [43–45]. The connection between the three strategies and the roles of the three strategies at different drought durations are still unknown.

Moso bamboo (*Phyllostachys edulis* (Carrière) J. Houz.) is one of the most important non-timber forest products and the fastest growing species in the world [46], which has great economic value and cultural significance [47,48]. Moso bamboo often dominates the competition with co-existing plants because of its strong resource competitiveness, which leads to the degradation of mixed forests into pure bamboo forests [49]. The strong resource competitiveness and high-speed growth of the aboveground part were attributed to the establishment and expansion of the huge root system during the growth of bamboo seedlings [50–52]. Seedling growth is a critical stage to explore the root formation strategy of plants because high mortality rates and high morphological plasticity under adverse conditions is often associated with the seedling phase [53]. At the seedling phase, the root is the main part of the development of Moso bamboo while the aboveground part grows slowly [52,54]. Therefore, seedlings of Moso bamboo were selected as experimental materials to explore drought adaptation strategies of roots during the seedling stage. Our study addresses the knowledge gap in adaptation strategies by documenting the ability of Moso bamboo seedlings to adjust root growth, root architecture, and NSC concentration in roots under drought with different intensities and durations. We aimed to (i) investigate the response patterns of Moso bamboo seedlings to drought with different intensities and durations, and (ii) explore the drought adaptation strategies of Moso bamboo root during the early stage of seedlings.

## 2. Materials and Methods

### 2.1. Experimental Set Up

This experiment was conducted in the Research Institute of Subtropical Forestry, Chinese Academy of Forestry (119°95′ E and 29°48′ N), which is located in Zhejiang province. The location belongs to the typical subtropical monsoon climate with a frost-free period of 307 d; and with mean annual sunshine hours of 1663.2 h, with mean June, July, August and annual temperatures of 24.1, 33.1, 33.4 and 17.8 °C, respectively; the average relative humidity is 70.3%. The average temperature during the test period was 25.30 °C. One kilogram of the soil (pH = 4.91) contained 18.7 g of organic carbon, 0.86 g of total N, 0.26 g of total P, 11.2 g of total K, 85.13 mg of hydrolyzable N, 4.15 mg of available P, and 65.73 mg of available K. Soil was derived from Moso bamboo forests in subtropical China to simulate natural conditions of bamboo growth. According to the distribution characteristic of Moso bamboo root, the soil for this experiment was excavated from an unfertilized 0–40 cm soil layer and mixed fully before potting.

Bamboo seeds were germinated on filter paper wetted with deionized water at 25 °C in an incubator until budding. Seedlings with similar length radicles were planted in a seedling disc. In April, after their leaves emerged, 90 bamboo seedlings with similar height (7.4 cm) and basal diameter (0.83 mm) were planted in plastic pots (25 cm × 27 cm). Each plastic pot was filled with 6 kg of soil. The soil moisture content was maintained at 80–85% of the maximum field water-holding capacity.

The drought experiment was conducted in a greenhouse from June 2017 to August 2017. Three drought levels were set in the study: the control (CK): approximately 80~85% of the maximum field water-holding capacity; moderate drought (M): 50~55%; and severe drought (S): 30~35%. The seedlings of the CK treatment grew well, indicating that they did not suffer any physical or nutritional limitation due to the substrate. To reduce the water content, the drought test group was allowed to dried naturally. The soil moisture

was controlled by combining the soil weighing method and a soil moisture measurement system (Aozuo ecological instrument, Trime-pico AZS-100, Beijing, China). First, the dry weight of 6 kg of soil was obtained by oven drying (105 °C). According to the water content of each treatment and the dry weight of soil, we calculated the total weight of pot culture to be reached for each treatment.

$$TWp = DWs (WCt + 1) + WP \tag{1}$$

where the TWp is total weight of pot culture, the DWs is dry weight of 6 kg soil, the WCt is water content of each treatment (CK, M and S), the WP is pot weight. The TWp of each treatment was a constant calculated by formula. Water was added to the pot until the total weight of the pot reached TWp, keeping the soil moisture content of each treatment within a predetermined range. Each treated soil was watered every 2 d. The soil of each treatment was watered once a day at 6 p.m.

### 2.2. Harvest and Measurements

Three sampling times were established. The first sampling (June) was carried out 30 d after the soil moisture content of each experimental group reached the expected level. Five seedlings were taken as five replicates per treatment. The interval of each sampling was 30 d, with the whole experimental period lasting for 90 d. We only monitored the root growth for three months because Moso bamboo seedlings can almost complete the root growth of one growing season within three months after planting.

The shoots were separated from the roots by a pair of scissors. Then, the soil was carefully shaken to avoid damaging the root tissue and root architecture, and the soil was subjected to a 2 mm sieve to obtain all residual roots. The roots and shoots were sealed in an ice box (0–2 °C) with a self-sealing bag and returned to the laboratory.

All roots were cleaned with clear water and scanned with a double-sided scanner at a resolution of 500 dpi (Regent Instruments Inc., WinRhizo Pro, Quebec City, QC, Canada). The root images were analyzed using WinRhizo software to obtain the root morphology parameters (root length, root surface, root average diameter, root tips and root links) and architecture parameters (FD and root branching angle (RBA)). The FD was obtained by WinRhizo software reference box-counting method [12,25]. Then, the roots, stems, and leaves were devitalized at 105 °C for 30 min to cease the their physiological activity and then dried at 65 °C to a constant weight to obtain the dry weight of roots and shoots (biomass) [55]. The SRL was calculated as root length divided by root dry weight. The concentrations of mobile sugars and starch in the roots were measured by anthrone colorimetry [34] after grinding the samples with a high-throughput tissue grinder (Retsch GmbH, MM400, Haan, Germany).

### 2.3. Calculations and Statistics

The effects of drought stress on root parameters could be quantitatively estimated using the effect size (EZ). Mt and Mc represent the value of the treatment group and the control group, respectively. The ratio of Mt to Mc was used as the response ratio (RR) of the roots to the treatment, and the natural logarithm of the RR was used to express the effect of the treatment on the roots. The logarithm of RR was taken to make the statistical test more convenient and to distinguish whether the effect of drought on the index was positive or negative [56].

$$EZ = \ln (RR) = \ln (Mt/Mc), \tag{2}$$

$$RLR = rl/RL. \tag{3}$$

The ratio of the root length of each diameter class (rl) to the total root (RL) length was used to calculate the root length ratio. Figure 1 denotes a classic classification of root topology.

$$TI = \lg (\alpha)/\lg (\mu) \tag{4}$$

where TI is the root topological index. The root tips and branch points are treated as external and internal nodes, respectively, and the root segments between the two nodes are referred to as links. Links that do not terminate in the organization are internal links, others are external links. The external links are classified as external–external (EE) when external links extend from other external links, external–internal (EI) when external links extend from internal links [11,25]. The altitude ($\alpha$) is the internal link number of the longest path from the root collar to an external tip, and the magnitude ($\mu$) is the total number of external links in the root system (total number of root tips) [15].

Differences in root traits among treatments and periods were tested using factorial analysis of variance (ANOVA). The correlations between NSC concentration and root architecture parameters (TI, FD, RBA) as well as SRL were analyzed by Pearson's correlation analysis. For all statistical tests, normality of residuals was assessed via Shapiro–Wilk test. SPSS statistical software was used for all analyses. Origin 9 and Excel 10 were used to construct all Figures and Tables.

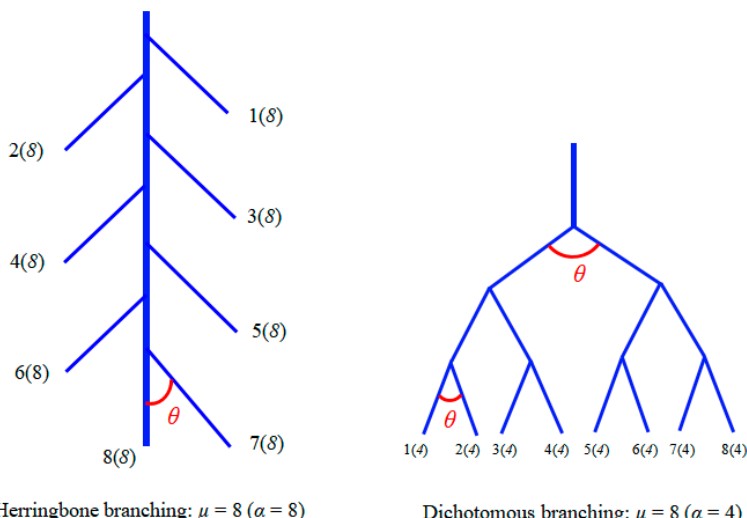

**Figure 1.** Schematic of root topology classification.

## 3. Result

### 3.1. Root Growth

Two-way variance analysis was performed to study the effects of two factors (treatments and periods) on the root morphological traits and the interaction between the two factors. Differences in root morphological indicators were significant between periods and between treatments ($p < 0.05$, Table 1). With respect to root length, root surface, root tips and root average diameter, significant interactions were observed between the two factors.

The accumulation of dry matter in the roots and shoots was significantly inhibited by treatments M and S. Within each treatment, the dry weight of the shoot and root showed a decreasing order as follows: CK > M > S for all the three periods. The root–shoot ratio of bamboo was increased by treatment M on the 30th day and 90th day and was decreased by treatment S on the 30th day and 60th day in comparison to the control (Figure 2).

Root length, surface area and root tips throughout the whole experimental period decreased significantly under drought treatment, and the inhibitory effect was enhanced with the deepening of drought stress (Figure 3a–c). The total root length, root surface area and root tips showed a decreased order of CK > M > S ($p < 0.05$). Root average diameter was not reduced until the 60th day (Figure 3d). The negative effect on the root surface area and root average diameter deepened significantly over time under the treatments M and S (Figure 3f–h), while that on root length and root tips did not deepen significantly during the third month (Figure 3e).

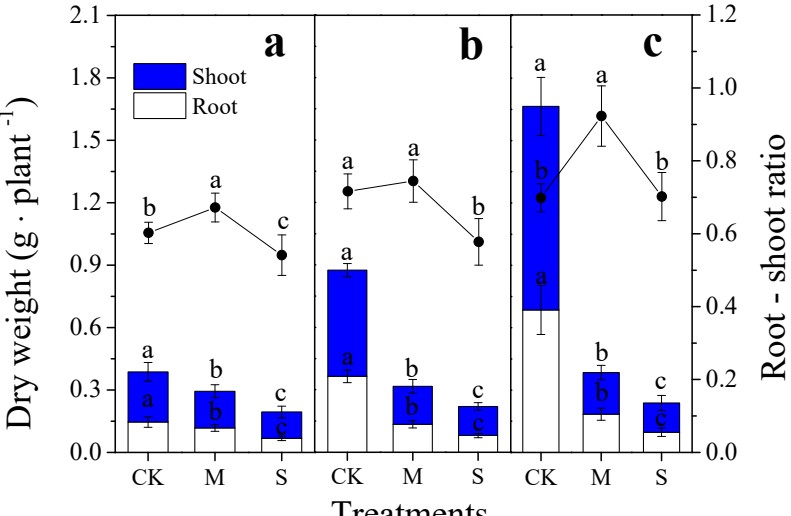

**Figure 2.** Effects of different drought treatments on the root–shoot ratio and on the dry weight of roots and shoots during different periods. The different lowercase letters denote significant differences between treatments ($p < 0.05$). The histogram represents the dry weight of the roots and shoots, and the line chart represents the root–shoot ratio. (**a**): 30th day; (**b**): 60th day; (**c**): 90th day. CK: control; M: moderate drought; S: severe drought. The same below.

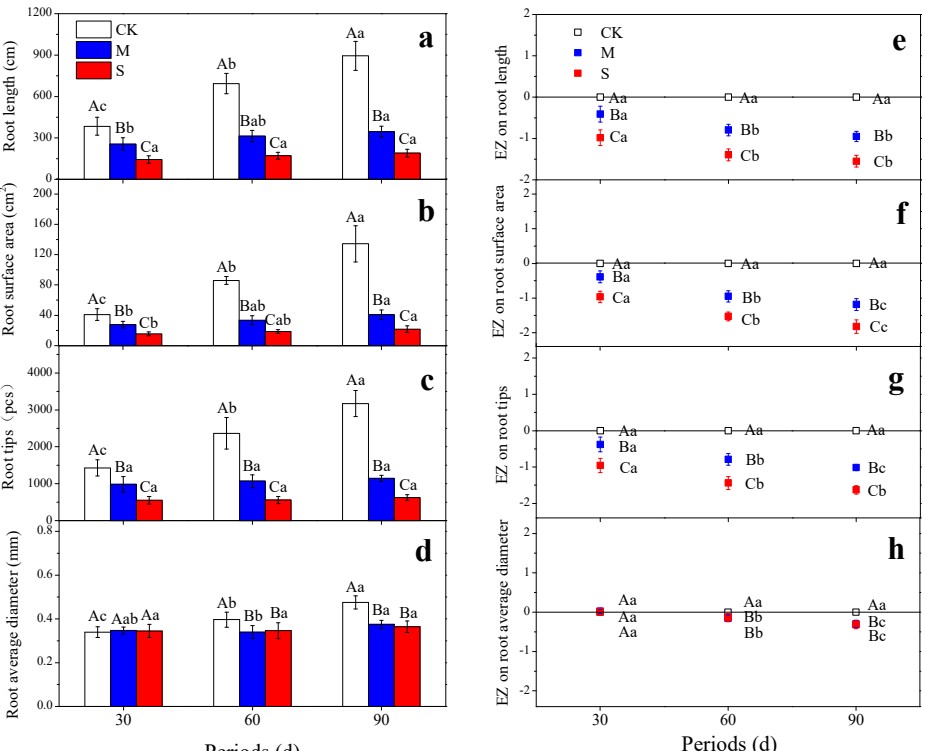

**Figure 3.** Effects of different drought treatments on root length (**a**), root surface area (**b**), root tips (**c**), root average diameter (**d**), EZ on root length (**e**), EZ on root surface area (**f**), EZ on root tips (**g**), and EZ on root average diameter (**h**) during different periods. The different capital letters denote significant differences between treatments ($p < 0.05$), and the different lowercase letters denote significant differences between periods ($p < 0.05$). The histograms represent root growth parameters, and the scatter diagrams represent the EZ of drought treatments on root growth. EZ: effect size.

**Table 1.** Effects of treatments, periods, and their interactions on plant growth and root architecture parameters (df = 44).

| Parameters | Treatment | | Period | | Treatment × Period | |
|---|---|---|---|---|---|---|
| | *F*-Value | *p*-Value | *F*-Value | *p*-Value | *F*-Value | *p*-Value |
| Root dry weight | 217.621 | <0.01 | 88.322 | <0.01 | 52.795 | <0.01 |
| Shoot dry weight | 278.269 | <0.01 | 81.496 | <0.01 | 70.227 | <0.01 |
| Root length | 307.212 | <0.01 | 56.829 | <0.01 | 26.772 | <0.01 |
| Root surface area | 224.849 | <0.01 | 61.557 | <0.01 | 34.116 | <0.01 |
| Root average diameter | 16.526 | <0.01 | 19.018 | <0.01 | 6.812 | <0.01 |
| Root tips | 236.522 | <0.01 | 31.731 | <0.01 | 21.624 | <0.01 |
| Fractal dimension | 27.981 | <0.01 | 25.392 | <0.01 | 0.496 | 0.739 |
| Specific root length | 1.548 | 0.227 | 17.951 | <0.01 | 7.433 | <0.01 |
| Root topological index | 16.743 | <0.01 | 0.621 | 0.543 | 1.723 | 0.166 |
| Root branching angle | 36.005 | <0.01 | 43.686 | <0.01 | 3.717 | 0.012 |

### 3.2. Root System Architecture

On the 30th day, the FD decreased significantly under the treatment S while the TI increased, and the EZ did not increase significantly over time. The effects of the treatment M on FD and TI were not significant until the 60th day, and the EZ did not change significantly over time (Figure 4a,b). The EZ of the treatment S on FD and TI reached their extremum value on the 60th day (Figure 4e,f). The average branching angle decreased significantly under the treatments M and S. The reduction effect gradually diminished over time (Figure 4c,g).

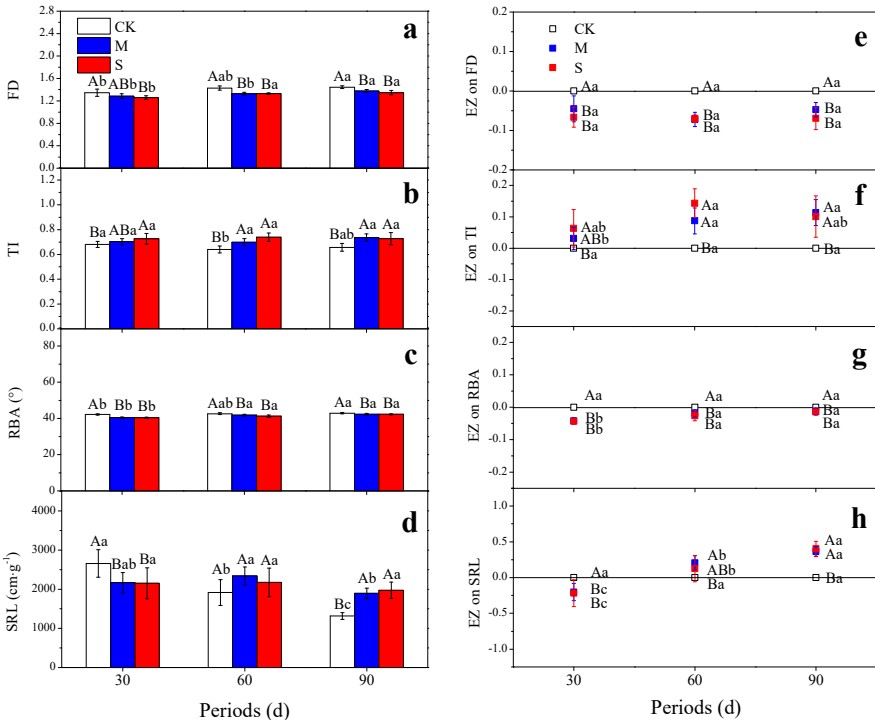

**Figure 4.** Effects of different drought treatments on fractal dimension (**a**), root topological index (**b**), root branching angle (**c**), specific root length (**d**), EZ on fractal dimension (**e**), EZ on root topological index (**f**), EZ on root branching angle (**g**), and EZ on specific root length (**h**) during different periods. The different capital letters denote significant differences between treatments (*p* < 0.05), and the different lowercase letters denote significant differences between periods (*p* < 0.05). EZ: effect size; FD: fractal dimension; TI: root topological index; RBA: root branching angle; SRL: specific root length. The histograms represent root architecture and SRL, and the scatter diagrams represent the EZ of drought treatments on root architecture and SRL.

### 3.3. Root Formation Strategy

The SRL can be used to judge the cost input of root growth and the root formation strategy [13,57]. On the 30th day, the SRL was significantly reduced by the treatments M and S. However, the negative effects of drought on the SRL gradually turned into positive effects over time. On the 90th day, within each treatment type, the SRL showed an increasing order as follows: S > M > CK (Figure 4d).

To gain insight into the causes of the differences in SRL and in root average diameter between the drought treatments and the control, we analyzed the root length distribution and the RLR of bamboo roots with different diameters under different treatments (Figure 5). On the 30th day, the roots (<0.3 mm, 0.7–0.8 mm, >0.9 mm in diameter) in treatment M were significantly shorter than in the control, while the roots in all diameter classes in treatment S were significantly shorter than in the control. However, the RLR of the roots with diameter ranging from 0.1 to 0.2 mm in treatment M was reduced in comparison to the control. In treatments M and S, the RLR of the roots with diameter ranging from 0.5 to 0.6 mm was increased (Figure 5a). On the 60th day, in treatments M and S, the RLR of the roots with diameter greater than 0.9 mm was reduced in comparison to the control (Figure 5b). On the 90th day, the effects of the two drought treatments on the length of roots in all diameter classes were similar to those on the 60th day. However, the RLR of roots with diameter greater than 0.8 mm was significantly reduced by the treatments M and S, while the RLR of roots with diameter ranging from 0.1 to 0.2 mm was significantly increased by treatment S (Figure 5c).

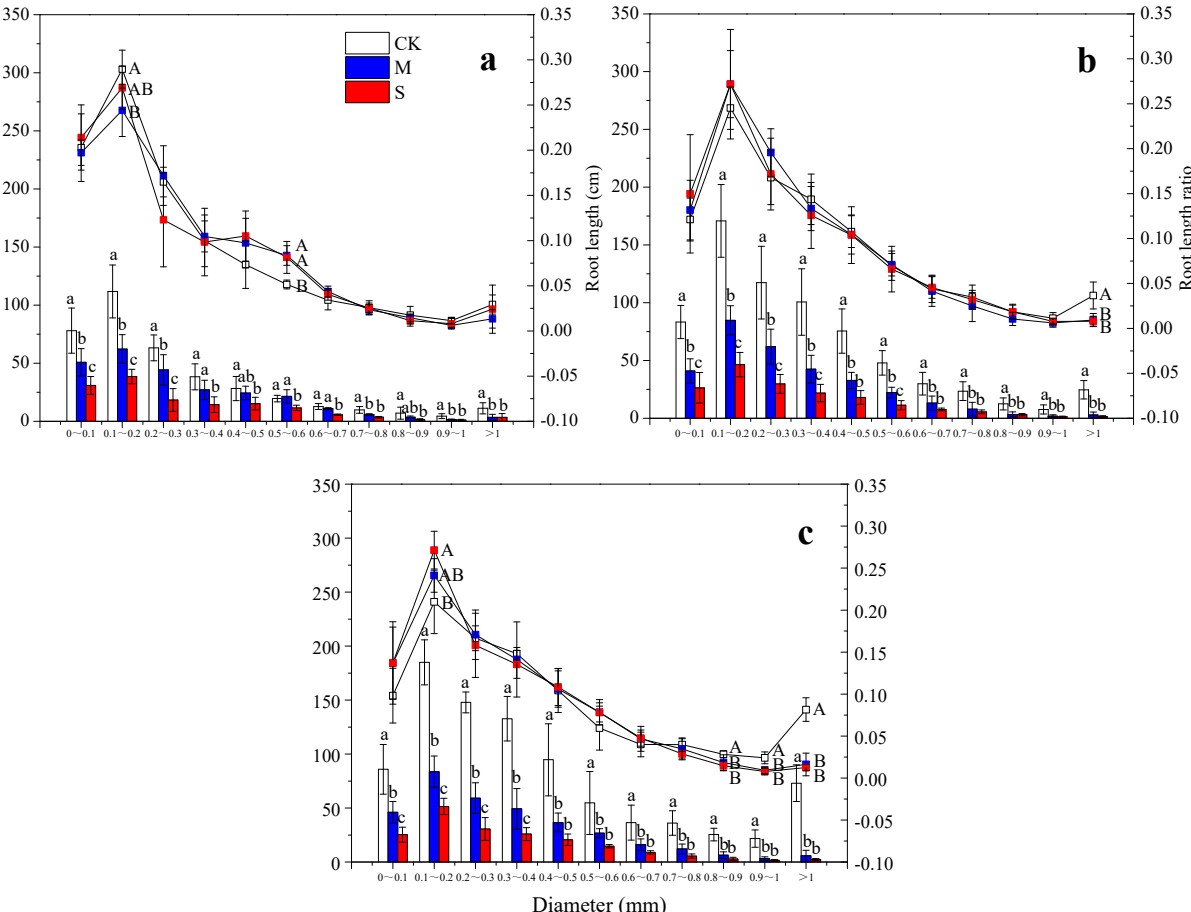

**Figure 5.** Root length distribution and the RLR of bamboo roots with different diameters under drought stress. The different capital letters denote significant differences in the RLR between treatments (*p* < 0.05), and the different lowercase letters denote significant differences in root length between treatments (*p* < 0.05). The histograms represent the root length, and the line charts represent the RLR. (**a**): 30th day; (**b**): 60th day; (**c**): 90th day.

### 3.4. Non-Structural Carbohydrates Response

The concentration of starch and NSCs in the root of the bamboo for three periods was significantly increased by drought stress (M and S). The effect of drought (M and S) on soluble sugar concentration was non-significant until the 60th day. The ratio of mobile sugars to starch was significantly reduced by the treatment M for three periods. The effect of the treatment S on the ratio of mobile sugars to starch was non-significant until the 60th day (Figure 6). Within each treatment type, the NSC concentrations showed a decreasing order as follows: S > M > CK.

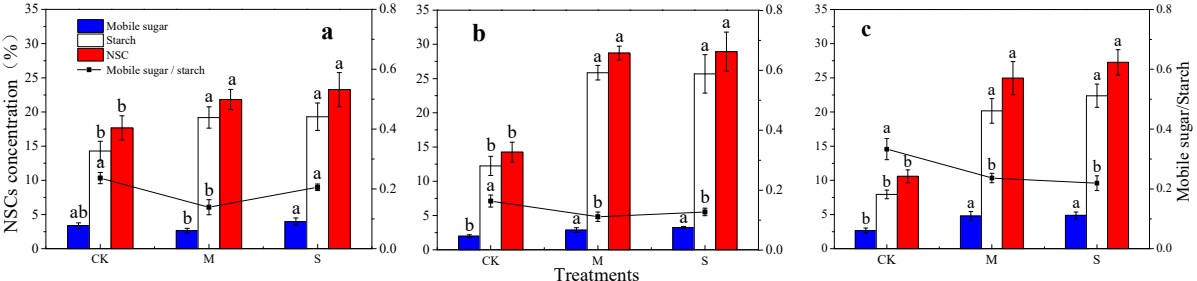

**Figure 6.** Effects of drought stress on the concentrations and composition of NSCs in bamboo roots. The different lowercase letters denote significant differences between treatments (*p* < 0.05). The histograms represent starch, mobile sugars or NSC, and the line charts represent the ratio of mobile sugars to starch. (**a**): 30th day; (**b**): 60th day; (**c**): 90th day. NSCs: non-structural carbohydrates.

### 3.5. Correlations between Root Morphology and NSC Concentration

The NSC concentration was strongly positively correlated with SRL and TI, but was strongly negatively correlated with RBA and FD, indicating that changes in root architecture and branching strength are closely related to changes in the NSC concentration in the roots (Figure 7).

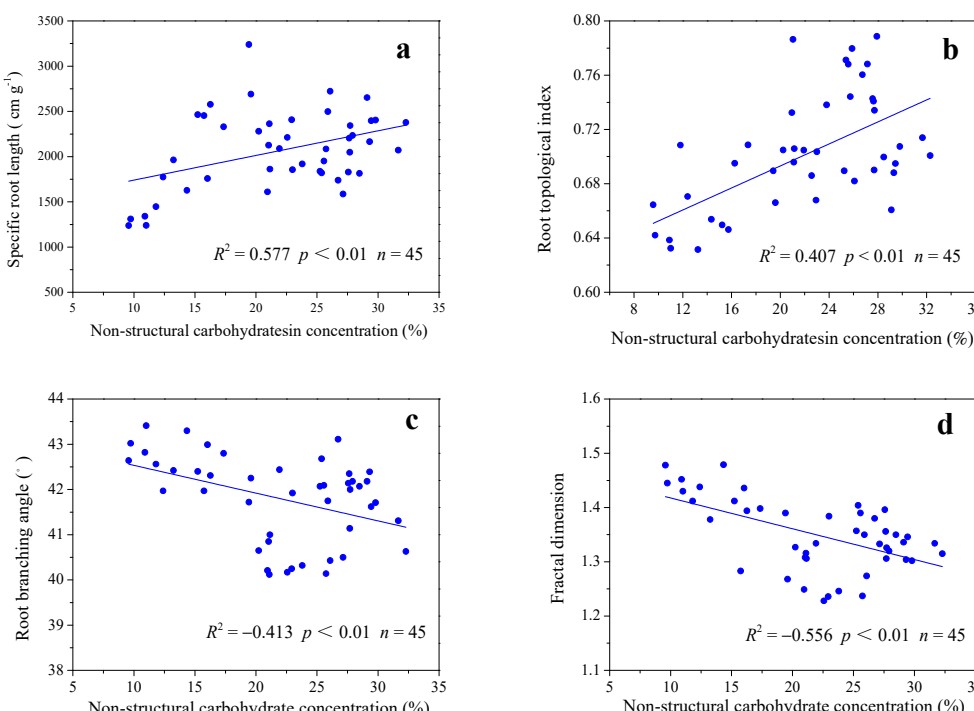

**Figure 7.** Relationships between non-structural carbohydrate concentration and root architecture parameters (root topological index (**b**), root branching angle (**c**), and fractal dimension (**d**)) as well as specific root length (**a**) during the whole experiment period. R: correlation coefficient; *n*: total number of samples.

## 4. Discussion

### 4.1. Root Morphology Response

In this study, the accumulation of dry matter was significantly inhibited by drought stress (Figure 2), which was consistent with the results of previous studies [58,59]. One recognized explanation for this phenomenon is that drought leads to a decrease in the rate of carbon dioxide assimilation and a reduction in photosynthesis products [60,61]. Another view is that the decrease of shoot growth reduced the water transpiration, resulting in moister and softer soil surrounding root tips, which was an advantage under drought conditions [58]. In addition, the decrease of root growth seems to help plants to resist drought stress, and it is not just a symptom of stress injury. It has been reported that the reduction of axial roots and lateral roots is beneficial to drought adaptation and deep soil moisture absorption [45]. The root–shoot ratio of bamboo increased in the treatment M (Figure 2), indicating that plants allocated a relatively high proportion of resources to the roots to improve the carbon use efficiency, similar to results in other species [62]. However, the trends of root–shoot ratio were not consistent between three periods, especially in the M treatment (Figure 2). A possible explanation is that root growth can conflict with shoot growth at a critical developmental state [63]. This conflict seems to be aggravated by the moderate drought.

On the 30th day, the SRL of the bamboo roots decreased in the treatments M and S but not for the average diameter (Figure 4d). A possible explanation is that drought stress forced the roots to accumulate a large amount of NSCs (Figure 6), thereby increasing the tissue density [14,62]. Surprisingly, during the last month of drought, bamboo roots showed another adaptation strategy where construction costs of roots were decreased by reducing diameter and increasing SRL (Figure 4d). Thinner roots require less carbon cost and lower respiratory metabolism and form a larger area of water absorption [13,64]. However, SRLs were observed increased [65] or stabilized [12,15] under drought conditions in previous studies. We speculate that sampling time and tree age may be the causes of this difference. The SRL of seedling roots seems to be more susceptible to change by fluctuant soil moisture than that of tree roots [14,15,66].

On the 30th day, the RLR of fine roots (with a 0.1–0.2 mm diameter) was significantly reduced by the treatment M (Figure 5), which may be ascribed to moderate drought inhibiting root proliferation (secondary roots) during the first month. Hanslin et al. reported that the development of secondary roots is easily inhibited in the early stages of drought because of its earliest development [64]. Effects of increased drought-induced soil impedance on root growth should also be taken into account. Experiments conducted with various soils clearly revealed that penetration resistance is directly correlated to bulk density, and it exhibits an inverse relationship to soil water content [67,68]. The decrease of very fine roots (0.1–0.2 mm) can increase the absorption and penetration ability of roots to dry soil [58,69]. Previous studies have shown that plants respond to the penetration resistance of the uppermost soil layer by reducing the number of lateral and axial roots [70,71], increasing root crown width [72] and root thickness [73], especially for the plants with fibrous roots, such as maize and wheat. Each newly initiated root needs to penetrate the uppermost soil layer to reach lower soil layers to acquire water from deeper soil. Furthermore, more coarse roots help maintain the stability of NSC pools under drought conditions [44].

On the 90th day, severe drought significantly increased the RLR of roots with a diameter of 0.1–0.2 mm, while the RLR of roots with a diameter greater than 0.8 mm decreased greatly in the two drought treatments (Figure 5), indicating that the radial growth of bamboo roots was significantly inhibited in the late stage of drought stress. Moso bamboo proliferated more fine roots with relatively less carbon to absorb more water, which can be regarded as the adaptation strategy of Moso bamboo seedlings in response to long-term drought. Drought has been found to stimulate fine root growth in maize [16]. However, contradictory results were found in drought studies of other species [17,73]. Our study surprisingly found that the inhibition effect of drought on thick roots or fine roots of bamboo was not fixed in different periods. Different adaptation strategies during

different periods are probably linked to the variation of the growth rate and water use efficiency in different periods [13,57]. Moso bamboo reduced the proliferation of fine roots in the early stages of drought to decrease respiratory metabolism, and later inhibited the thickening of roots to decrease the construction cost of roots. Allometric growth characteristics of most plants may eventually lead to different adaptation strategies when plants are subjected to drought stress at different growth stages [74,75]. In addition, several studies on root function showed that roots absorb water mainly through low-order fine roots (<0.5 mm in diameter) with a high surface area-to-volume ratio, while water transport occurs mainly through high-order roots (>0.5 mm in diameter) with well-developed xylem conduits [14,76,77]. Therefore, our results suggest that drought affects the proliferation of absorptive roots in the first 30 d, and then has a relatively greater negative impact on the formation of transport roots.

### 4.2. Root Architecture Response

Moso bamboo seedlings reduced the competition within the root system and increased the costs of root building by increasing their TI, thereby improving their adaptability to drought conditions (Figure 4b). Hanslin et al. found that the TI of two species increased after 15 d of drought stress [65]. According to the characteristics of the TI and the water absorption capacity of different tree species [25,45], the strategy of increasing the TI in arid soil is helpful to absorb water. However, the basic root characteristics of bamboo, similar to the dichotomous branching type, did not change under the drought condition (Figure 4b). This is consistent with the results of a previous study by Alvarez-Flores [15]. According to the results of the TI, the negative effect of drought on the lateral root branching is stronger than the growth of the primary root, which is supported by Lynch's result that a root system with fewer axial roots is more advantageous under drought [45]. Plants with more developed primary roots and deeper roots have been reported to have relatively higher water absorption efficiency and superior drought adaptability [28]. In addition, the change in the TI represents not only the change in root branch pattern, but also the adjustment of resource allocation mode. The TI directly reflects the carbon input of the root system and the absorption efficiency of water and nutrients [78]. Another advantage of increasing the TI of bamboo roots is to optimize carbon allocation, which is based on lateral roots being considered expensive in terms of root respiration [79,80]. Gao and Lynch observed that plants lessen root internal competition and devote more resources (photosynthates and water) to the formation of taproots and deeper roots by reducing lateral root branches [81].

The RBA can reflect the direction of growth and water-seeking (horizontal or longitudinal) of roots under drought conditions. In this study, the branching angle decreased gradually with increasing degree of drought stress, which is a useful strategy (Figure 4c). It has been reported that the ideal root system with "steeper and deeper" architecture had a relatively higher water absorption efficiency [21,82]. A similar evidence was observed by Manschadi, who found that drought-tolerant wheat genotypes exhibited a narrower RBA in comparison to drought-sensitive wheat genotypes [28]. This method of reducing the RBA to avoid root formation in shallow soil layers may be related to the gradual soil moisture distribution [20,21,83]. However, the response of plants to drought by reducing the RBA has not been reported as an adaptation strategy to drought in previous studies.

The FD of the bamboo roots reduced significantly in the treatments M and S (Figure 4d), indicating that drought inhibited the roots' development and reduced the roots' system internal competition by reducing the roots' complex branches. Reducing the growth of lateral roots is also a drought resistance strategy. The study of Zhan observed that reducing the lateral root branching density improves drought tolerance of maize [84]. Similarly, van Oosterom et al. suggested that the parsimonious root has more advantages in drought soil in comparison to a complex root [85]. However, after the 30th day, the EZ of drought on the FD did not increase over time, suggesting that the effect of drought on the density and angle of root branching occurred in the early stage of drought (Figure 5). The increase of root TI and the decrease of root branch density (FD) in bamboo during the first 60 d of

drought can also be confirmed by the decrease of RLR of fine roots (0.1–0.2 mm) (Figure 5). The branch strength decreased due to the decrease of fine root proliferation, which is also the reason for the decrease of FD and the increase of the TI in the early stage of drought. The adjustment of root architecture occurred in the early stage of seedling growth, which may be related to the formation of lateral roots being inhibited by drought or soil impedance. This is an innovative conclusion and can be used to improve root architecture to adapt to drought conditions. For example, re-watering after a particular drought time may be a better irrigation model, since a short-term drought can make Moso bamboo form more drought-tolerant and water-absorbent root architecture (higher TI and lower RBA).

Changes in root architecture also appear to be associated with the increase of penetration resistance due to drought [68,86]. Studies of soil resistance have found that drought caused plants to distribute more roots in the upper soil layer [72,87,88]. Bécel et al. suggested that the influence of soil penetration resistance on root branch density or topology may be more significant than that on root elongation, and speculated that penetration resistance caused plants to develop dichotomous roots [68]. The opposite conclusion was drawn after topological and fractal analysis of the root system of Moso bamboo seedlings. The decrease in lateral root elongation and branching due to high penetration resistance [89] may eventually lead to the increase in the TI and the decrease in FD of Moso bamboo in the early stages of seedling development. The adjustment of RBA and TI of Moso bamboo indicates that the root system tends to develop deeper and steeper herringbone roots. However, a larger proportion of roots were assigned to the upper soil layer due to the limitation of soil resistance, which leads to the decrease of water absorption space and the total root amount. This may be the reason why the biomass of Moso bamboo is still severely reduced, although it can form a more drought-tolerant root system under drought conditions. The monitoring of soil impedance and the details of the relationship between soil impedance and root growth in this study are insufficient, and will be improved in the future research.

### 4.3. Adaptation Strategy Related to Carbon Investment

In this study, the concentrations of NSCs in bamboo roots significantly increased in the treatments M and S (Figure 6), which may be deemed to be one of the adaptation strategies (increase carbon investment) of bamboo. Changes in NSC concentrations under drought are not uniform, concentrations may increase or decrease or become stable, depending on the tolerance of plants to drought [7,90,91]. The change in NSC concentrations in plant tissues reflects not only the physiological response to environmental changes but also the balance between carbon used for structural growth and respiratory metabolism [37,92]. Under drought conditions, drought-adapted trees tend to keep a high NSC concentrations in their roots to maintain growth and tissue osmotic pressure [93–95], while less-adapted trees often reduce the NSC concentration to maintain root respiration [90,96]. Since drought inhibits plant growth earlier than photosynthesis, NSC concentrations may increase in the early phases of drought [77], but NSC concentrations may also decrease during longer drought because compounds are constantly consumed by respiratory metabolism and osmotic adjustments [33]. Compared with fine roots, the concentrations of NSCs in relatively thick roots under drought stress are more susceptible to the carbon balance mentioned above, while NSCs in fine roots are often used to regulate water potential [95,97].

Our result showed that there was a significant correlation between NSC concentration and root architecture parameters (Figure 7), indicating that the sensitivity of NSC concentration to drought, to some extent, supported the root architecture plasticity. It is generally recognized that the reduction of lateral roots causes an increase in TI, resulting in a decrease in SRL [45,58]. The reduction of lateral root branches and the lateral growth with the increase of the NSC concentration is supported by Kannenberg et al., who found a potential trade-off of allocation photoassimilates between NSCs and biomass under dry conditions; for example, lateral root reduction will be accompanied by an increase in NSC accumulation [44]. Importantly, NSC concentration and SRL are positively corre-

lated (Figure 7a), indicating that plants may construct low-cost roots with more carbon investment. In addition, studies have shown that certain plant species balance deficiencies in tissue radial growth by increasing NSC concentrations within the growth parts when adapting to drought stress [43,44]. The NSC concentration in the bamboo root was negatively correlated with the RBA (Figure 7c). Therefore, we proposed that the decrease in the RBA is related to the geotropism enhancement resulting from the increased starch concentrations under drought conditions [98–100]. From the perspective of carbon investment, the seedlings of Moso bamboo maintained high investment of carbon throughout the experimental period to support the plasticity of root architecture and morphology under drought conditions. According to the construction process of the root system, the high-cost root architecture was built in the early stage of seedling growth and the low-cost root morphology was built in the later stage. This is a very meaningful conclusion that can be considered as a theoretical basis for improving the drought resistance of seedlings. For example, measures such as elevated $CO_2$ [101], temperature adjustment [102] and elevated ozone [103] that increase the NSC concentration in roots may be beneficial to plant resistance to drought, since increasing the NSC concentration can support the root architecture plasticity of seedlings.

The strong resource competitiveness of adult Moso bamboo is not only due to the strategy of root formation at the seedling stage, but also related to the high plasticity of the rhizome system and the physiological integration mechanism of cloned plants [50,104]. The rhizome response strategies of Moso bamboo of different ages to drought should be studied emphatically in the future research. In this study, a drought experiment of container seedlings was used to simulate the natural soil environment of drought to analyze the root adaptation strategy of Moso bamboo seedlings. However, the experimental conditions did not fully illustrate the impact of all rainfall events on Moso bamboo in nature. Therefore, watering episodes of different characteristics should be taken into account in future studies, such as intense rainy events occurring in a short period of time, long rainy periods of low intensity or re-watering after drought. In addition, soil drought is not a single stress. Soil impedance, heat and hypoxia caused by drought have been proved to affect root morphogenesis [63,68]. Therefore, variation factors associated with drought will be considered to investigate the response mechanism of root morphology in future research.

## 5. Conclusions

Moso bamboo seedlings showed different root adaptation strategies in response to drought with different intensities and durations. Moso bamboo seedlings formed a "steeper, simpler, expensive (low SRL and high TI)" root architecture to adapt to a short-term drought (one month) by decreasing the SRL, FD and RBA and increasing the TI. Subsequently, Moso bamboo seedlings formed a "cheaper (high SRL)" root to adapt to the carbon scarcity caused by long-term drought (three months). Moso bamboo seedlings always maintain high carbon investment in root systems by increasing the NSC concentration in the root under drought conditions, thus supporting the plasticity of root architecture and morphology (i.e., the TI, FD, RBA and SRL). Root adaptation strategies of Moso bamboo seedlings for drought are not isolated, but support each other to form a drought resistance system.

The early stage of seedling growth (one month) is the key phase for root architecture construction. Short-term drought stress at the early phase of seedling growth or measures that increase the NSC concentration in roots may be beneficial to plants' resistance to drought by optimizing the root architecture. This finding is of great significance in constructing the root architecture models with superior water absorption capability and higher carbon utilization efficiency. This study revealed the water foraging strategy of bamboo seedlings under drought conditions, thereby improving the understanding of the water competition model of Moso bamboo and helping to optimize soil water management technology in the Moso bamboo forest.

**Author Contributions:** Conceptualization Z.Y. and B.Z.; Formal Analysis Z.Y., Y.C. and J.Z.; Investigation Z.Y., X.G., Q.L. and M.L.; Writing—Original Draft Z.Y. and M.L.; Writing—Review and Editing B.Z. All authors have read and agreed to the published version of the manuscript.

**Funding:** This study was supported by funding from the National Natural Science Foundation of China (NSFC) (Nos. 31670607, 30840064), National Key R&D Program of China (No. 2016YFD0600202-4) and the Fundamental Research Funds for the Central Non-profit Research Institution of CAF (Nos. CAFYBB2017ZX002-2, CAFYBB2018GB001).

**Acknowledgments:** We are sincerely grateful to the anonymous reviewers and editors for their valuable suggestions to improve the article.

**Conflicts of Interest:** The authors declare no conflict of interest.

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
