# Peer review of "Root Response of Moso Bamboo (Phyllostachys edulis (Carrière) J. Houz.) Seedlings to Drought with Different Intensities and Durations"

_forests, doi:10.3390/f12010050_

Round 1
Reviewer 1 Report
The manuscript is well written. Just a few comments.
Line 128: Soil chemical conditions are given. How deep was soil excavated in forest or which soil layer(layers) was used for you pot experiment?
Line 171 “Then, the roots, stems, and leaves were devitalized at 105 ℃ for 30 minutes to minimize the physiological activity [34].”
- Why this reference is used? Can´t see the relation.
- Mai-He Li, Yong Jiang, Ao Wang, Xiaobin Li, Wanze Zhu, Cai-Feng Yan, Zhong Du, Zheng Shi, Jingpin Lei, Leonie Schönbeck, Peng He, Fei-Hai Yu, Xue Wang, Active summer carbon storage for winter persistence in trees at the cold alpine treeline, Tree Physiology, Volume 38, Issue 9, September 2018, Pages 1345–1355, https://doi.org/10.1093/treephys/tpy020
- Isn´t 60 ℃ enough to minimize physiological activity?
Line 441 Correct reference is “Kannenberg et al.” or “Kannenberg et al. (2017)”
Author Response
Comment 1: Line 128: Soil chemical conditions are given. How deep was soil excavated in forest or which soil layer(layers) was used for you pot experiment?
Response: Thank you for your suggestion. Per your suggestion, relevant soil information was added to the revised manuscript. See line 131-133.
“ According to the distribution characteristic of moso bamboo root, the soil used in this experiment was excavated from unfertilized 0 - 40 cm soil layer and mixed fully before potting. ”
Comment 2: Line 171 “Then, the roots, stems, and leaves were devitalized at 105 ℃ for 30 minutes to minimize the physiological activity [34].”
- Why this reference is used? Can´t see the relation.
- Mai-He Li, Yong Jiang, Ao Wang, Xiaobin Li, Wanze Zhu, Cai-Feng Yan, Zhong Du, Zheng Shi, Jingpin Lei, Leonie Schönbeck, Peng He, Fei-Hai Yu, Xue Wang, Active summer carbon storage for winter persistence in trees at the cold alpine treeline, Tree Physiology, Volume 38, Issue 9, September 2018, Pages 1345–1355, https://doi.org/10.1093/treephys/tpy020
- Isn´t 60 ℃ enough to minimize physiological activity?
Response: (1) Per your suggestion, the error has been modified. We have added the correct reference. See line 172-174.
- We have corrected the misunderstand caused by our uncleardescription. Drying at 105 ℃ is a common method to inactivate plants and cease their internal physiology activity (https://doi.org/10.1016/j.catena.2019.104094; DOI: 10.1007/s11104-013-1965-9).
Comment 3: Line 441 Correct reference is “Kannenberg et al.” or “Kannenberg et al. (2017)”
Response: Per your suggestion, “Kannenberg” was replaced by “Kannenberg et al.”. See line 445.

Reviewer 2 Report
The paper brings new insights into the response of moso bamboo roots to the intensity and duration of drought in young seedlings. All chapters in the paper are well structured and written and I especially commend the rich list of literature and discussion. The paper provides concrete insights into what happens to the morphology of the root system and its physiological role in the preservation of carbohydrates and starches during dry periods. The results can be used in nurseries that produce bamboo seedlings to optimally irrigate the optimal morphology, architecture, and physiological role of roots. Seedlings grown in this way show optimal biomass ratios of the aboveground and underground part and are under less physiological shock, which is reflected in this balance of transpiration and adsorption. The work is also useful in establishing new moso bamboo plantations to determine in advance the soil moisture conditions that are optimal for its growth and growth and ultimately the production of biomass and carbon dioxide sinking which contributes to a better quality of life (economic importance), especially their survival in rural areas and in reducing global and local climate change.
The introduction is thorough and well written. There are no objections to the materials and methods of work. It may be possible to display some data in the results in tabular form. The pictures are graphically and statistically good devices but difficult to follow, especially I mean Figure 5. I commend the rich discussion with numerous quotes and comparisons with other species on similar topics. The bibliography is impressive.

Author Response
Thanks to reviewer 2 for your very meaningful comments. We have revised the article and responded to all comments.
Comment 1: Add author (Carrière) J. Houz.
Response: Per your suggestion, “(Carrière) J. Houz.” was added to the Latin name of moso bamboo. See line 2, 19, 104.
Comment 2: Add the unit of root tips.
Response: Per your suggestion, “pcs” was added to figure 3. See line 232.
Comment 3: Delete authors.
Response: Per your suggestion, the error has been corrected. See line 304.
Comment 4: It may be possible to display some data in the results in tabular form. The pictures are graphically and statistically good devices but difficult to follow, especially I mean Figure 5.
Response: Thank you for your suggestion. We modified the three graphs to attempt to clarify them. We finally decided to show the data in graph form, because we want to show the length distribution trend of root with different diameters more intuitively. See line 257-259.
